# Socio-medical factors associated with neurodevelopmental disorders on the Kenyan coast

Patricia Kipkemoi[1,2,3]*, Jeanne E. Savage[2], Joseph Gona[1], Kenneth Rimba[1], Martha Kombe[1], Paul Mwangi[1], Collins Kipkoech[1], Eunice Chepkemoi[1], Alfred Ngombo[1], Beatrice Mkubwa[3], Constance Rehema[1], Symon M. Kariuki[1,4,5], Danielle Posthuma[2,6], Kirsten A. Donald[7,8], Elise Robinson[9,10], Amina Abubakar[1,3,4,5], Charles R. Newton[1,3,4,5]

**1** Neuroscience Unit, KEMRI-Wellcome Trust Research Programme, Kilifi, Kenya, **2** Complex Trait Genetics Department, Center for Neurogenomics and Cognitive Research (CNCR) Vrije Universiteit Amsterdam, Amsterdam, The Netherlands, **3** Institute for Human Development, Aga Khan University, Nairobi, Kenya, **4** Department of Psychiatry, University of Oxford, Warneford Hospital, Oxford, United Kingdom, **5** Department of Public Health, Pwani University, Kilifi, Kenya, **6** Department of Child and Adolescent Psychology and Psychiatry, Complex Trait Genetics, Amsterdam University Medical Centres, Vrije Universiteit Amsterdam, Amsterdam Neuroscience, Amsterdam, The Netherlands, **7** Department of Paediatrics and Child Health, Red Cross War Memorial Children's Hospital and University of Cape Town, Rondebosch, South Africa, **8** Neuroscience Institute, University of Cape Town, Groote Schuur Hospital, Cape Town, South Africa, **9** The Broad Institute of MIT and Harvard, Cambridge, Massachusetts, United States of America, **10** Center for Genomic Medicine, Massachusetts General Hospital, Boston, Massachusetts, United States of America

* patricia.kipkemoi@aku.edu

## Abstract

Neurodevelopmental disorders (NDDs) are a group of conditions with their onset during the early developmental period and include conditions such as autism and intellectual disability. Occurrence of NDDs is thought to be determined by both genetic and environmental factors, but data on the role of environmental factors for NDD in Africa is limited. This study investigates environmental influences on NDDs in children from Kenya. This case-control study compared children with NDDs and typically developing children from two studies on the Kenyan coast. We included 172 study participants from the Kilifi Autism study and 151 from the NeuroDev study who had a diagnosis of at least one NDD and 112 and 73 with no NDD diagnosis from each study, respectively. Potential risk factors were identified using unadjusted univariable analysis and adjusted multivariable logistic regression. Univariable analysis in the Kilifi Autism study sample revealed hypoxic-ischaemic encephalopathy conferred the largest odds ratio (OR) 10.52 [95%CI: 4.04, 27.41] for NDDs, followed by medical complications during pregnancy (gestational hypertension & diabetes, eclampsia, maternal bleeding) (OR=3.17 [95%CI: 1.61, 6.23]). In the NeuroDev study sample, labour and birth complications (OR=7.30 [95%CI 2.17, 24.61]), neonatal jaundice (OR=5.49 [95%CI 1.61,18.72]) and infection during pregnancy (OR= 5.31 [95%CI 1.56, 18.11]) conferred the largest risk associated with NDDs. In the adjusted

**Data availability statement:** Coded individual-level data that do not allow researchers to identify participants will be made available by the authors, without undue reservation. Curated data for this manuscript, has been deposited to the Harvard Dataverse: https://doi.org/10.7910/DVN/KTIOB1.

**Funding:** The Kilifi Autism study was funded through the Cheryl & Reece Scott Professorship Award to Prof. Charles Newton. NeuroDev is supported by the Stanley Center for Psychiatric Research at the Broad Institute, a grant from SFARI (704413), and by the National Institute of Mental Health of the National Institutes of Health under Award Number U01MH119689. Research reported in this publication was also supported by the Eunice Kennedy Shriver National Institute of Child Health & Human Development of the National Institutes of Health under Award Number R01HD102975.

**Competing interests:** At the time of data collection, JG, KR, and AA were supported by the Cheryl & Reece Scott Professorship Award to CRN at the KEMRI-Wellcome Trust Research Programme. PK, MK, PM, CK, EC, AN, BM, and CR are fully supported by the NeuroDev Study in the course of this study. AA, CRN, KAD, and ER report grants from the Stanley Center for Psychiatric Research at the Broad Institute and the National Institute of Health during the conduct of this study. JES and DP report support from grants operated by the Complex Trait Genetics Department, Vrije Universiteit Amsterdam, Netherlands. The authors declare no competing interests.

analysis, seizures before age 3 years in the Kilifi Autism study and labour and birth complications in the NeuroDev study conferred the largest increased risk. Higher parity, the child being older and delivery at home were associated with a reduced risk for NDDs. Recognition of important risk factors such as labour and birth complications could guide preventative interventions, developmental screening of at-risk children and monitoring progress of these children. Further studies examining the aetiology of NDDs in population-based samples, including investigating the interaction between genetic and environmental factors, are needed.

## Introduction

Neurodevelopmental disorders (NDDs) are a range of conditions affecting cognitive and behavioural development, with notable examples being autism and intellectual disability [1]. Most NDD studies have been carried out in high-income settings, with a recent review of global autism prevalence only including data from three African countries and other Africa-based studies being smaller-scale case-control studies [2]. Research indicates that the prevalence of NDDs, such as intellectual disability and autism, in parts of Africa and other lower-middle-income countries (LMICs) may be higher due to the incidence of risk factors for NDDs. These include limited access to healthcare facilities and undernutrition and other risk factors for neuro-behavioural disorders, such as adverse perinatal events, infections of the brain and environmental toxins, which are more prevalent in these regions [3,4].

NDDs are a complex group of conditions with environmental and genetic risk factors implicated in its aetiology [5–7]. Studies show that there are critical periods of neurodevelopment, where exposure to adverse environmental factors, such as viral or bacterial infection of the pregnant mother, exposure to specific medications and chemical and physical pollutants in the prenatal period, can disrupt typical neurobiological development, and these disruptions may be associated with NDDs [8–10]. Environmental factors such as mode of delivery and birth weight have been associated with the risk of NDDs, emphasising the multifactorial nature of these conditions [11]. Additionally, advanced maternal age, low maternal education, maternal alcohol and tobacco use, gestational diabetes, and hypertension have been linked to an increased risk of intellectual disability [11]. Studies have found prenatal factors, including uterine bleeding, certain medications during pregnancy, low birth weight, and preterm delivery, to be associated with autism [8]. Maternal immune activation and elevated cytokines have been linked to an increased risk of NDDs [12]. These prenatal factors are more common in LMICs; for example, preterm births, South Asia and Africa contribute to 80% of the global burden [13]. This has changed little in the last decade [14]. Evidence from population-based studies in the Global North has identified the role of sociomedical risk factors in the aetiology of autism [15]. There is limited evidence from African countries [4], with one study by Mankoski et al. found evidence of falciparum malaria as a possible antecedent to autism in Tanzania [16]. The response to infections, such as malaria, may be an important contribution

to this risk during pregnancy that warrants further investigation. Widely researched risk factors associated with NDDs, as reviewed by Guinchat et al., such as pre-eclampsia, placental insufficiency, prolonged labour, induced labour, birth asphyxia, preterm birth and low birth weight, are common in Africa [17]. Bleeding and maternal infection during pregnancy have also been linked to NDDs. There has been strong epidemiological evidence linking advanced parental age, both maternal and paternal, to an increased risk for autism [18]. Biological evidence, however, on paternal-age-related *de novo* variants and the associated risk with autism and other conditions has found small causal effects [19]. There is also evidence that maternal age effects on *de novo* variants are small [20]. Lack of access to healthcare facilities leads to poor maternal health status, which in turn can lead to higher incidences of NDDs [21].

Historically, research on NDDs has focused on the Global North; environmental factors such as infectious diseases during pregnancy and limited healthcare resources during pregnancy would allow the investigation of risk and protective factors specific to Africa. These could potentially inform public health strategies towards the early identification of autism in these settings. This study aims to identify prenatal, perinatal and postnatal factors associated with NDDs, such as autism and intellectual disability in Coastal Kenya, comparing children with autism to children with other NDDs as well as typically developing children.

## Methods

### Ethics statement

The Kilifi Autism study was approved in August 2012 by the Kenya Medical Research Institute National Ethics and Review Committee (reference number: KEMRI/RES/7/3/1). For the NeuroDev study, ethical approval was sought in Kilifi, Kenya, and approval was granted by the Kenya Medical Research Institute Scientific Ethics and Review Unit (KEMRI/SERU/CGMR-C/104/3629) in April 2018 and the Harvard T.H. Chan School of Public Health IRB17-0600 in June 2018. Written consent was obtained from the parents or caregivers of the child participants.

### Research design and setting

The study examines data from two datasets: an ongoing case-control study that aims to characterise the genetics and phenotypic architecture of NDDs in children (NeuroDev study [22]) and a study aimed at validating an autism-specific screening tool and, after that, collecting data on risk factors associated with autism (Kilifi Autism Study). The participants from the Kilifi Autism study were recruited from mainstream schools, special needs units, and special needs schools in Kilifi and Mombasa counties in Kenya between 5th October 2012 and 20th September 2013. NeuroDev Kenya participants included in this analysis were recruited from previous studies in the neuro assessment department, specialised clinics, and special schools in Kilifi County. The study began data collection on 13th February 2019 and is ongoing, this analysis includes a subset of data collected up to March 2020, when the study paused its collection due to the COVID-19 pandemic.

### Procedures and measures

Data collection for the Kilifi study is concluded, while the NeuroDev study collection is ongoing with one-point data collection for both studies. Information about the study was shared with potential participants in the language the participants were most comfortable in (Kiswahili or Kigiryama). Informed written consent to participate in the study was sought from all parents or caregivers of children with or without NDDs enrolled on the two studies. Verbal assent was requested from all child participants, with written assent requested for all children above 13 years. With the two datasets, we approached the analysis with a discovery and replication approach. When we compared the selection process and the demographic and clinical characteristics of the participants, there were enough similarities in the variables of interest to evaluate these two datasets using this approach.

**Kilifi Autism study.** Eligible parents and children were recruited from mainstream schools, special needs units and special schools after extensive community engagement efforts with the relevant stakeholders in the health and education ministries. Typically developing children and children with a presumptive diagnosis of NDD (autism, severe learning disabilities and intellectual disability) from the Educational Assessment Resource Centre (EARC) were identified in the special schools. We did not administer an IQ test to confirm cognitive ability. As such, we use here the presumptive diagnosis for intellectual disability from the EARC.

A trained fieldworker shared information about the study and sought informed written consent to participate. A positive screen on the social communications questionnaire (SCQ) or an endorsement of autism from the ADOS or the DSM criteria was used to define autism case status. Those with the presumptive EARC diagnosis of intellectual disability or severe learning disabilities and no autism diagnosis, as confirmed by the SCQ to the ID/NDD group. These study measures were translated into the local languages, Kiswahili and Kigiriyama, through a standardised forward and back translation process (Table 1). A panel/team involved in translations included a developmental psychologist and trained professionals (linguists and research assistants) who were fluent in English, Swahili, and Kigiriyama and familiar with the local culture.

**NeuroDev study.** Children with an NDD and affected siblings included in the NeuroDev study had clinical diagnoses of a neurodevelopmental disorder based on the DSM-5 criteria or a presumptive diagnosis from EARC, were within the specified age range (2–17 years old) and were willing to participate. As with the Kilifi Autism study, we did not administer an IQ test before enrolment into the study. As such, we used the presumptive diagnosis of intellectual disability from the EARC.

Participants were recruited from previous studies carried out in the Neuroscience unit in Kilifi and from mainstream and special needs schools on the Kenyan coast. We included participants with a diagnosis of at least one NDD and controls with no diagnosis of an NDD in this current study. Children with NDDs were excluded if they had a co-occurring primary neuro-motor condition such as moderate to severe cerebral palsy. The exclusion of children with a co-occurring primary neuro-motor condition, an indication that the child was not able to walk, was added as some of the assessments such as the Molteno Adapted Scales and the neuro-medical assessment included an examination of gross motor development,

**Table 1. Standardised tools for the Kilifi Autism study.**

| Measure | Domain | Administration | Psychometric properties | Variables |
|---|---|---|---|---|
| Socio-demographic questionnaire (Neuro-assessment department, bespoke) [23] | Family socio-economic status | To parent or care-giver | N/A | Child and parent age, Sex, Ethnicity, Education, NDD Diagnosis |
| Neuromedical questionnaire [24,25] | Birth history, child neurology | To parent or care-giver | N/A | Birth weight, Number of children ever born, Pregnancy medical complications, Infection or fever during pregnancy, Gestational term, Labour and birth complications, No cry at birth, difficulties breathing and breastfeeding, delivery place, seizures in the family (first and second-degree relatives), jaundice in the first 30 days of life, malaria before age 3 years |
| Social communication questionnaire [26] | Autism screen | To parent | Internal consistency - Cronbach alpha = 0.90; Confirmatory factor analysis of the 3 Factor DSM-IV-TR model = root mean squared error of approximation (RMSEA) = 0.050; Comparative Fit Index (CFI) = 0.974; Tucker-Lewis Index (TLI) = 0.973 | NDD diagnosis |

which would not be able to be assessed. Controls were included in the study if they did not have a diagnosis of a neuro-developmental disorder, were within the study age range and were matched according to catchment area, ancestry and age.

The study collects data on demographics and socio-economic status through the use of the Kilifi Asset Index [27]. A specialised neuromedical questionnaire was used to collect a range of clinical features, including birth history, family history, growth, neurological conditions, and medical conditions (Table 2). Measures such as the Developmental Diagnostic Dimensional Interview (3Di), Swanson, Nolan and Pelham (SNAP)-ADHD rating scale and the Ravens progressive matrices were used to evaluate autistic traits, ADHD traits, and non-verbal reasoning, respectively. These cognitive and behavioural measures were not used in this analysis.

## Participants

The Kilifi Autism study sample included 268 children; 167 had a neurodevelopmental concern from teacher and caregiver reports, further delineated as the autism group (n = 78) and the NDD group (n = 89) after administration of the Autism Diagnostic Observation Schedule (ADOS), and 101 were reported to be typically developing. Autism and intellectual disability co-occurred in some children (ID) (n = 54, 19%).

Table 2. Standardised tools for the NeuroDev study.

| Tool | Domain | Administration | Psychometric properties | Variables |
|------|--------|----------------|-------------------------|-----------|
| Intake and Demographics questionnaire, bespoke [28,29] | Sociodemographic | To parent | N/A | Child and parent age, Sex, Ethnicity, NDD Diagnosis |
| Raven's Progressive Matrices [30] | Child and parental cognition | To child and parent 5 - 90 years | Internal consistency - Cronbach alpha = 0.81 | N/A |
| Molteno Developmental Scale [31] | Child cognition | To child 2–5 years | Correlation with the Bayles Scales of Infant Development Pearson's $r$ = 0.70) | N/A |
| 3Di Short Version [32] | Child autism traits | To parent | Internal consistency – McDonald's omega = 0.83; sensitivity [66.7% (95% CI: 0.22–0.96)] specificity [82.5% (95% CI: 0.74–0.89)] | N/A |
| SNAP-IV-ADHD [33] | Child ADHD traits | To parent | Internal consistency – Cronbach alpha = 0.90; Correlation with the Child Behaviour Checklist - Pearson's $r$ = 0.55) | N/A |
| Childhood Behavioral Checklist (CBCL), pre-school and school versions [34] | Child ADHD traits | To parent | Internal consistency – Cronbach alpha = 0.95; Test-retest reliability- Pearson's $r$ = 0.76 | N/A |
| UCT Red Cross Hospital Neuromedical Assessment, adapted | Birth history, child neurology | To parent & child | N/A | Birth weight, Number of children ever born, Pregnancy medical complications, Infection or fever during pregnancy, Gestational term, Labour and birth complications, No cry at birth, difficulties breathing and breastfeeding, delivery place, seizures in the family (first and second-degree relatives), jaundice in the first 30 days of life, cerebral malaria in childhood (<6 years), meningitis in childhood |
| Kilifi Asset Index [27] | Family socioeconomic status | To parent | | Education status |

The NeuroDev study analysis sample consists of 72 children in the autism group, 93 in the NDD (ID without autism) group and 72 in the typically developing group that are clinically diagnosed. In the NeuroDev Kilifi cohort, we noted that autism co-occurs with ID, with few participants having a diagnosis of autism alone; as such, we combined the diagnostic groups for analysis into children with an NDD vs typically developing children. While there are strong health and education resources available to tease apart severe global developmental delays and intellectual disability, there are still a lot of resources needed to strengthen these systems in the identification of autistic individuals with low or moderate support needs [35]. This finding is similar to others on the African continent, such as clinical settings in Nigeria, where more severe presentations of autism are seen with an overrepresentation of non-verbal children as well as co-occurring intellectual disability and ADHD [36,37]. Prenatal factors such as severe maternal obesity are associated with autism (with or without ID) [38]. A study by Schieve and colleagues found that perinatal factors such as preterm birth, low birth weight and low APGAR scores were associated with autism, ID and autism with co-occurring ID [39]. There is plausible biological and epidemiological evidence that supports the grouping of autism and intellectual disability as conditions on the neurodevelopmental continuum, as discussed in the Research Domain Criteria and adds credence to the investigation of risk factors of these two conditions grouped as NDDs [40].

## Data analyses

**Data preparation.** We analysed the risk factors that have been noted as relevant and linked to autism in previous studies. To reduce the number of tests we carried out, we grouped some variables into larger categories. For example, maternal infection was operationalised as a mother reporting a fever (yes/no) during pregnancy; for labour and birth complications, this was operationalised as 'Were there emergencies or problems during the delivery?' (yes/no) in the NeuroDev study, and if the mother/caregiver recalled this, we noted the problem; these included complications such as induced labour and prolonged labour, major obstetric haemorrhage, premature rupture of membranes (PROM), and umbilical cord complications. In the Kilifi Autism study, this was operationalised in the following questions: 'Difficult or prolonged labour'. Parents/caregivers also noted any abnormality in the antenatal and delivery periods. Hypoxic-ischaemic encephalopathy (HIE) is operationalised in this study as no cry at birth for the child, difficulty and needing assistance to breathe after birth, and not being able to suckle normally. Due to the limited recording of birth details in medical records in many rural health facilities, we are not able to use Appearance Pulse Grimace Activity Respiration (APGAR) scores.

A number of parental variables had < 10% of missingness, with paternal age at the time of birth having the highest missingness (62% in the Kilifi Autism study and 38% in the NeuroDev study). Maternal age at the time of birth was second highest in terms of missingness, with 24% in the Autism study and 11% in the NeuroDev study. Paternal education was also highly missing in the Kilifi Autism study, at 24%.

**Data analysis.** Descriptive statistics were computed to describe the study sample according to diagnosis status (Table 3). We presented the frequencies and proportions for categorical variables, mean ± SD for continuous parametric data and median and first and third quartile for non-parametric continuous data. We compared the distribution of sociodemographic and medical characteristics of NDD vs typically developing children using chi-square tests (or Fisher's exact test if the frequency was ≤ 5) for categorical variables and non-parametric tests such as Mann-Whitney U test were used on the raw scores of continuous variables to evaluate the difference in proportions for the study variables. The Kilifi Autism study, with a sample size of 286 participants, was adequately powered to detect large differences (Cohen's d = 0.9) in proportions between groups with 100% power for chi-square tests and 99% for the Mann-Whitney U tests. The NeuroDev study, with a sample size of 224 participants, was adequately powered to detect large differences (Cohen's d = 0.9) in proportions between groups with 100% power for chi-square tests and 96% for the Mann-Whitney U tests.

**Table 3. Autism and NeuroDev Study Participant Characteristics.**

| Participant characteristics and socio-demographic data | Kilifi Autism study | | | NeuroDev Study | | |
|---|---|---|---|---|---|---|
| | NDDs (n=173) | TD (n=113) | NDD vs TD p-value | NDDs (n=151) | TD (n=73) | NDD vs TD p-value |
| **Child Sex Male** | 107 (61.9%) | 66 (58.4%) | 0.560[c] | 94 (62.3%) | 37 (50.7%) | 0.100[c] |
| **Child age in years at assessment (Median (Q1, Q3))** | 12 (9, 15) | 9 (6, 11) | 0.316[b] | 12 (10, 14) | 11 (10, 12) | 0.448[b] |
| **Mother's age in years (Median (Q1, Q3))** | 37 (32, 42) | 34 (29, 40) | 0.376[b] | 37 (32, 46) | 40 (33, 45) | 0.530[b] |
| **Maternal age at birth in years (Median (Q1, Q3))** | 25 (21, 31) | 25 (21, 30) | 0.473[b] | 26 (22, 33) | 29 (23, 34) | 0.560[b] |
| <30 | 91 (52.6%) | 58 (51.3%) | 0.408[c] | 82 (54.3%) | 29 (39.7%) | **0.007**[c] |
| 30-34 | 28 (16.2%) | 12 (10.6%) | | 32 (21.2%) | 16 (21.9%) | |
| >35 | 17 (9.8%) | 11 (9.7%) | | 27 (17.9%) | 12 (16.4%) | |
| Missing/Can't recall | 37 (21.4%) | 32 (28.3%) | | 10 (6.6%) | 16 (21.9%) | |
| **Father's age in years (Median (Q1, Q3))** | 45 (39, 52) | 42 (36, 52) | 0.490[b] | 46.5 (40.0, 55.5) | 46 (42, 54) | 0.489[b] |
| **Paternal age at birth in years (Median (Q1, Q3))** | 33.5 (29, 40) | 31.5 (27.0, 41.0) | 0.499[b] | 35 (29, 43) | 35 (30, 43) | 0.502[b] |
| <30 | 21 (7.3%) | 10 (8.9%) | **0.024**[a] | 28 18.5%) | 6 (8.2%) | **0.026**[a] |
| 30-34 | 25 (8.7%) | 8 (7.1%) | | 20 (13.3%) | 9 (12.3%) | |
| >35 | 9 (3.2%) | 12 (10.5%) | | 54 (35.8%) | 20 (26.2%) | |
| Missing/Can't recall | 95 (33.2%) | 83 (73.5%) | | 49 (32.5%) | 38 (52.1%) | |
| **Parental age gap in years: Median (Q1, Q3)** | 8 (3,11) | 7.5 (4.5, 11.5) | 0.544[b] | 7 (4, 11) | 7 (4.5, 8.5) | 0.477[b] |
| **Maternal education** | | | | | | |
| Never attended | 44 (25.4%) | 60 (53.1%) | **<0.001**[a] | 57 (37.8%) | 38 (52.1%) | **0.018**[a] |
| Primary | 70 (40.5%) | 35 (31.0%) | | 66 (43.7%) | 33 (45.2%) | |
| Secondary | 30 (7.3%) | 3 (2.7%) | | 18 (11.9%) | 2 (2.7%) | |
| Tertiary | 7 (4.1%) | 0 (0.0%) | | 5 (3.3%) | 0 (0.0%) | |
| Missing/Can't recall | 22 (12.7%) | 15 (13.3%) | | 5 (3.3%) | 0 (0.0%) | |
| **Paternal education** | | | | | | |
| Never attended | 16 (9.3%) | 13 (11.5%) | **0.013**[a] | 20 (14.0%) | 12 (37.5%) | 0.350[a] |
| Primary | 62 (38.8%) | 61 (54.0%) | | 74 (51.8%) | 44 (60.3%) | |
| Secondary | 34 (7.5%) | 13 (11.5%) | | 31 (21.7%) | 12 (16.4%) | |
| Tertiary | 13 (5.9%) | 3 (2.7%) | | 10 (7.0%) | 1 (1.4%) | |
| Missing/Can't recall | 48 (27.8%) | 23 (20.4%) | | 8 (5.6%) | 4 (5.5%) | |
| **Ethnicity** | | | | | | |
| Giriama | 65 (37.7%) | 49 (43.4%) | **<0.001**[a] | 80 (53.0%) | 50 (68.5%) | **0.005**[a] |
| Chonyi | 16 (9.3%) | 21 (18.6%) | | 25 (16.6%0 | 17 (23.3%) | |
| Kauma | 13 (7.5%) | 10 (8.85%) | | 3 (2.0%) | 3 (4.1%) | |
| Other Mijikenda | 10 (5.8%) | 12 (10.6%) | | 17 (11.3%) | 3 (4.1%) | |
| Luo | 6 (3.5%) | 0 (0.0%) | | 6 (4.0%) | 0 (0.0%) | |
| Other | 41 (23.7) | 6 (5.3%) | | 15 (9.9%) | 1 (1.4%) | |
| **Missing** | 22 (12.7%) | 15 (13.3%) | | 5 (3.3%) | 0 (0.0%) | |
| **Number of children ever born: Median (Q1, Q3)** | 5 (3, 7) | 6 (5, 8) | 0.638[b] | 5.9 (2.66) | 7.4 (2.3) | **<0.001**[b] |
| **Birth order: Median (Q1, Q3)** | 3 (2, 5) | 4.0 (2.0, 5.5) | 0.596[b] | 3.5 (2, 6) | 5 (3, 7) | 0.620[b] |
| **Medical complications during pregnancy (gestational hypertension, diabetes, eclampsia and maternal bleeding)** | 50 (32.7%) | 13 (13.3%) | **0.001**[c] | 11 (7.3%) | 2 (2.7%) | 0.230[c] |

*(Continued)*

**Table 3.** (Continued)

| | Kilifi Autism study | | | NeuroDev Study | | |
|---|---|---|---|---|---|---|
| Infection during pregnancy (fever, malaria and other infections) | 36 (23.7%) | 20 (23.8%) | 1.000[c] | 28 (18.4%) | 3 (4.1%) | **0.003**[c] |
| Drug misuse during pregnancy | 17 (10.9%) | 16 (17.9%) | 0.124[c] | 14 (9.3%) | 3 (4.1%) | 0.281[c] |
| Delivery place – home | 91 (58.7%) | 79 (46.5%) | **<0.001**[c] | 85 (56.7%) | 63 (42.6%) | **<0.001**[c] |
| Labour and birth complications (induced labour and prolonged labour, Premature rupture of membranes (PROM), major obstetric haemorrhage (MOH), umbilical cord complications and meconium aspiration) | 29 (18.7%) | 4 (4.0%) | **<0.001**[c] | 36 (17.4%) | 3 (4.1%) | **<0.001**[c] |
| Hypoxic ischaemic encephalopathy (HIE) | 56 (36.1%) | 5 (5.1%) | **<0.001**[c] | 25 (16.6%) | 3 (4.1%) | **0.001**[c] |
| Birth weight in kgs (Mean, SD) | 3.4 (1.16) | 3.7 (0.68) | 0.618[c] | 3.1 (2.5, 3.5) | 3.5 (3.0, 3.6) | 0.631[c] |
| Low birth weight (≤ 2.5kg) | 27 (15.6%) | 4 (3.5%) | **<0.001**[c] | 23 (15.2%) | 1 (1.4%) | **0.002**[c] |
| Previous hospitalisation in childhood | 94 (60.7%) | 19 (19.4%) | **<0.001**[c] | 47 (31.1%) | 10 (13.7%) | **0.005**[c] |
| Family history of seizures | 15 (9.7%) | 12 (12.2%) | 0.538[c] | 34 (26.0%) | 15 (22.1%) | 0.605[c] |
| Neonatal jaundice | 15 (13.3%) | 1 (0.9%) | 0.410[c] | 28 (18.7%) | 3 (4.1%) | **0.002**[c] |
| Seizures at birth | 9 (5.2%) | 1 (0.9%) | **0.047**[c] | 23 (15.2%) | 3 (4.1%) | **0.017**[c] |
| Head injury or coma | 16 (10.4%) | 3 (3.1%) | **0.048**[c] | Not assessed | | |
| Malaria before 3 years | 8 (4.6%) | 4 (.5%) | 0.450[a] | Not assessed | | |
| Seizures before 3 years | 32 (18.5%) | 2 (1.8%) | **<0.001**[a] | Not assessed | | |
| Family history of NDDs | Not assessed | | | 15 (9.9%) | 0 (0.0%) | **0.002**[c] |
| Cerebral malaria anytime in childhood | Not assessed | | | 16 (10.5%) | 1 (1.4%) | **0.014**[c] |

Note: NDD = Neurodevelopmental Disorder, TD = Typically Developing CI = Confidence Interval, Q1, Q3 = Quartile 1, Quartile 3, SD = standard deviation (mean and SD are provided for continuous variables with a normal distribution and median and Q1, Q3 are provided for count variables or continuous variables without normal distribution), p-values in **bold** <0.05,

[a] = Pearson's chi-squared test (dichotomous and categorical variables,

[b] = Mann Whitney U test;

[c] = Fisher's exact test,

[d] = t-test on raw data (continuous variables).

Our outcome (NDD diagnosis) is dichotomous; as such, we carried out unadjusted analysis of the potential risk factors using logistic regression of the risk factors of interest and calculated odds ratios (OR) and 95% confidence intervals (95%CI). We also computed the risk ratios (RR) [41]. Risk factors with a p-value reaching 0.25 in the univariate analysis were included in the multivariable analysis. Multivariable analysis was used to assess the association between the identified risk factors and NDDs, calculating odds ratios and 95%CI. For the multivariable analysis, parental and child characteristics such as age, parental education level, and ethnicity were added *a priori* into the multivariable model as covariates to account for any potential confounding of the other risk factors. Sociodemographic factors such as parental education have been strongly linked to child health and development outcomes [42,43]. Studies in the United States, India and China have found that higher parental education is associated with an earlier autism diagnosis [44–46]. Both the Kilifi Autism and NeuroDev studies were adequately powered to detect large associations (Cohen's d = 0.9) between the risk factors and NDDs, with a *post hoc* power calculation of 100% for both studies.

To address the issue of multiple comparisons in our study examining potential risk factors associated with autism, we employed the Benjamini-Hochberg Procedure to control the False Discovery Rate (FDR) [47]. This method is particularly suitable for situations where many hypothesis tests are conducted simultaneously, as in this study with 35 tests. After performing the necessary statistical tests to assess the association between each potential risk factor and the outcome of

interest in the unadjusted univariable analysis, we obtained a set of p-values representing the significance of these associations. Specifically, we sorted the obtained p-values in ascending order and calculated the critical value corresponding to our desired FDR level (typically set at 0.05). Then, we compared each p-value to its corresponding critical value and considered it significant if it fell below this threshold.

Analyses were conducted in STATA version 15.0 (StataCorp LP, College Station, Texas, United States of America [USA]) and used package LOGITTORISK to compute risk ratios and the package METAN to estimate the effect size and standard error.

## Results

### Descriptive characteristics of the study participants

**Kilifi Autism study.** The final sample from the Kilifi Autism study included 173 children with an NDD diagnosis and 113 typically developing children (controls) with a median age of 10 years (7,13). More than half of the participants were male (61.9%). There were no statistically significant differences in child sex and age between the diagnostic groups.

The median ages of the mothers of NDD children (37 years) were not statistically different (p = 0.376). Maternal age at delivery was not different between the groups. 36.4% of mothers and 10.1% of fathers did not have formal schooling. Maternal education levels were statistically significantly higher in the NDD group compared to the typically developing groups (p < 0.001). There were significant differences in prenatal and perinatal factors between the NDD and neurotypical groups. For example, medical complications during pregnancy, such as gestational hypertension, gestational diabetes, eclampsia and maternal bleeding, were reported more in mothers of children with NDD (32.7%) as compared to mothers of neurotypical children (13.3%), and this difference was statistically significant (p < 0.001).

**NeuroDev study.** The NeuroDev study included 151 children with an NDD and 73 typically developing controls with a median age of 11 years (9,13) (Table 3). There were more male children in the study (62.3%). The age of the mothers in the NDD group was 26 years old, and mothers of controls had a median age of 29 years. The difference in the ages, both currently and at birth, is not statistically different (p = 0.560). 14.3% of fathers and 42.4% of mothers did not have formal schooling. Maternal education levels were statistically significantly higher in the NDD group compared to the typically developing groups as well (p = 0.018).

Groups did not differ in terms of most demographic characteristics, such as the father's and mother's age at assessment, paternal education levels and paternal age at delivery. Infection during pregnancy (fever, malaria) was more common in the NDD group (18.4% v. 4.1%, p = 0.003) compared to the neurotypical group. Labour and birth complications were also reported more in mothers of children with NDDs (17.4% v. 4.1%, p < 0.001), similar to HIE (16.6% v. 4.1%, p < 0.001), low birth weight (<2.5kgs) (15.2% v. 1.4%, p < 0.002).

### Risk factors comparison between NDD and Typically developing groups: Univariable analysis

Many of the factors conferred an increased risk for NDDs, with odds ratio (OR) ranging from 1.05 to 12.60 in the Kilifi Autism study and 2.30 to 7.30 in the NeuroDev study. Higher parity (more children born to a mother) (OR=0.84 [95%CI 0.77, 0.93]; OR=0.78 [0.68, 0.90]) and decreasing birth order (OR=0.91 [0.82, 0.99]; OR=0.85 [0.75, 0.95]) showed less risk for NDD in both studies). In a curious finding, 'Delivery at home' conferred the least risk for NDDs in both studies (OR=0.32 [95%CI 0.18, 0.59] and (OR=0.21 [95%CI 0.10, 0.44). To explore this result further, we controlled for maternal education here, and we still found low risk with an OR of 0.28 [95%CI 0.13, 0.70] and 0.24 [95%CI 0.11, 0.51] in the Kilifi Autism and NeuroDev studies, respectively. We adjusted here for pregnancy and labour complications, and we see this effect persists in the NeuroDev study with an OR of 0.28 [95%CI 0.13, 0.65] but is non-significant in the Kilifi Autism

study with an OR of 0.57 [95%CI 0.29, 1.12). We pooled the ORs from both studies and computed a meta-analysed OR, also in Table 4 below.

### Risk factors comparison between NDD and Typically developing groups: Multivariable analysis

Factors that reached the p-value threshold of ≤ 0.25 in the multivariable analysis, highlighted with the superscript [m], were selected for the multivariable model, and we adjusted for the following sociodemographic factors: maternal and paternal education and ethnicity. Ten factors reached the statistically significant threshold in the Kilifi Autism study, and seven factors reached the statistically significant threshold with the adjusted multivariable analysis (S1 Table).

**Child and parent factors.** Mother's age at the time of assessment was associated with NDDs (OR= 1.05 [95%CI 1.01, 1.09]) in the Kilifi Autism study. The number of children born (parity) was non-significant in the Kilifi Autism study (OR=0.96 [95%CI 0.86, 1.08]) but significant in the NeuroDev study (OR=0.84 [95%CI 0.71, 0.98)]. Birth order was significant in the Kilifi Autism study (OR=0.87 [95%CI 0.01, 0.97]) but non-significant in the NeuroDev study (OR=0.90 [95%CI 0.80, 1.02]). Child male sex was non-significant in the NeuroDev study after the adjusted multivariable analysis (OR=1.62 [95%CI 0.89, 2.94], p = 0.115). Parental age gap was non-significant in the Kilifi Autism study (OR=0.97 [95%CI 0.91, 1.03]).

**Pre-natal factors.** Medical complications in pregnancy (gestational hypertension & diabetes, eclampsia, and maternal bleeding) were significant in the Kilifi Autism study (OR=2.73 [95%CI 1.31, 5.69]). Infection during pregnancy was significant in the NeuroDev study (OR=4.27 [95%CI 1.20, 15.16]). Drug misuse during pregnancy was significant in both studies, however interestingly in different directions - lower risk in the Kilifi Autism study (OR=0.45 [95%CI 0.20, 0.98]) and increased risk in the NeuroDev study (OR=4.12 [95%CI 1.08, 15.75]). Labour and birth complications were non-significant in the Kilifi Autism study (OR=2.83 [95%CI 0.89, 9.01) but were associated with NDDs in the NeuroDev study (OR=6.32 [95%CI 1.81, 22.03]). Delivery at home was non-significant in the Kilifi Autism study (OR=0.56 [95%CI 0.28, 1.10]) and associated with a low risk for NDDs in the NeuroDev study (OR=0.29 [95%CI 0.13, 0.64]).

**Perinatal factors.** HIE was associated with NDDs in the Kilifi Autism study (OR= 9.54 (95%CI 3.51, 25.97]) but was non-significant in the NeuroDev study (OR=1.00 [95%CI 1.00 – 1.01]). Birth weight was significant and associated with a low risk for NDDs in the Kilifi Autism (OR=0.70 [95%CI 0.49, 0.98]) and NeuroDev OR=0.46 [95%CI 0.23, 0.99]) studies. There was no effect noted for low birth weight (<2.5kgs) in the Kilifi Autism study (OR=1.00 [95%CI 0.99, 1.00) and was non-significant in the NeuroDev study OR=1.00 [95%CI 0.99, 1.00]). Seizures at birth were non-significant in both studies.

**Neonatal factors.** Neonatal jaundice in the NeuroDev study (OR = 6.56 [95%CI 1.85, 23.26]) and seizures before age 3 in the Kilifi Autism study (OR=13.00 [95%CI 3.02, 55.90]) were significant factors associated with NDDs. Cerebral malaria in childhood (<6 years) was non-significant in the NeuroDev study OR=7.57 [95%CI 0.95, 60.21]), and head injury in childhood was non-significant in the Kilifi Autism study (OR=2.97 [95%CI 0.77, 11.46]).

## Discussion

This study investigated the association of specific prenatal, perinatal and childhood factors associated with autism and intellectual disability on the Kenyan coast. We found that adverse prenatal, perinatal and childhood events were more common in children with NDDs compared to typically developing children but that some of these were significant in the discovery data (Kilifi Autism study) and not in the replication dataset (NeuroDev study). This may be related to a number of factors including that these datasets were collected at different time periods when the distributions may have differed, and differences in methodological approaches in data collection scales used. These differences were reflected in the descriptive analysis that showed distribution differences between NDD and TD groups for variables such as higher levels of education status of mothers for the Kilifi Autism study and reducing proportion of older maternal age in the NeuroDev cohorts. These differences, among others, may explain the findings that some associations with NDD were found in one dataset but not the other.

**Table 4. Unadjusted analysis of relevant parental, perinatal and neonatal factors associated with autism – NDD vs Typically Developing.**

| Risk factor variables | Kilifi Autism study | | | | NeuroDev Study | | | | OR Meta-analysis from the two studies |
|---|---|---|---|---|---|---|---|---|---|
| | Odds ratio [95% CI] | p-value | Adjusted p-value | Risk Ratio [95%CI] | Odds ratio [95% CI] | p-value | Adjusted p-value | Risk Ratio [95%CI] | |
| Child Male sex | 1.15 [0.71, 1.87] | 0.561 | 0.686 | 1.13 [0.73,1.72] | 1.60 [0.91, 2.82] | 0.101 | 0.214[m] | 1.51 [0.92, 2.39] | 1.34 [0.03, 2.65] |
| Child age in years at assessment Median (Q1, Q3) | **1.17 [1.10, 1.25]** | **0.001** | **0.004[m]** | 1.15 [1.09, 1.22] | 1.05 [0.95, 1.16] | 0.311 | 0.491 | 1.05 [0.96, 1.14] | 1.11 [0.02, 2.19] |
| Mother's age in years Median (Q1, Q3) | **1.05 [1.02, 1.09]** | **0.005** | **0.015[m]** | 1.05 [1.02, 1.08] | 0.99 [0.95, 1.02] | 0.484 | 0.631 | 0.99 [0.96, 1.02] | 1.02 [0.02, 2.02] |
| Maternal age at birth in years Median (Q1, Q3) | 1.02 [0.98, 1.06] | 0.433 | 0.621 | 1.02 [0.08,1.05] | 0.98 [0.94, 1.02] | 0.232 | 0.387 | 0.98 [0.95, 1.02] | 1.00 [0.02, 1.98] |
| <30 | Reference group | | | | Reference group | | | | |
| 30-34 | 1.49 [0.70, 3.16] | 0.301 | 0.452 | 1.42 [0.72,2.60] | 0.71 [0.34, 1.47] | 0.355 | 0.507 | 0.73 [0.36, 1.40] | 0.96 [0.02, 1.90] |
| >35 | 0.99 [0.43, 2.25] | 0.971 | 0.983 | 0.99 [0.46, 2.00] | 0.80 [0.36, 1.77] | 0.576 | 0.665 | 0.82 [0.39, 1.64] | 0.88 [0.02, 1.90] |
| Father's age in years Median (Q1, Q3) | 0.99 [0.95, 1.03] | 0.537 | 0.682 | 0.99 [0.96, 1.03] | 1.00 [0.96, 1.04] | 0.960 | 0.982 | 1.00 [0.96, 1.04] | 0.99 [0.02, 1.97] |
| Paternal age at birth in years Median (Q1, Q3) | 0.98 [0.94, 1.02] | 0.259 | 0.407 | 0.98 [0.95, 1.02] | 1.00 [0.97, 1.02] | 0.732 | 0.784 | 1.00 [0.97, 1.02] | 0.99 [0.02, 1.96] |
| <30 | Reference group | | | | Reference group | | | | |
| 30-34 | 1.49 [0.50, 4.45] | 0.477 | 0.630 | 1.42 [0.53, 3.31] | 0.33 [0.10, 1.12] | 0.077 | 0.178 | 0.35 [0.11, 1.11] | 0.54 [0.01, 1.07] |
| >35 | 1.15 [0.42, 3.16] | 0.785 | 0.836 | 1.13 [0.45, 2.56] | 0.78 [0.26, 2.35] | 0.658 | 0.757 | 0.80 [0.28, 2.07] | 0.93 [0.02, 1.84] |
| Parental age gap in years: Median (Q1, Q3) | 0.96 [0.91, 1.01] | 0.092 | 0.169[m] | 0.96 [0.92, 1.01] | 1.02 [0.96, 1.08] | 0.567 | 0.665 | 1.02 [0.96, 1.07] | 0.99 [0.02, 1.96] |
| Maternal education[m] | | | | | | | | | |
| Never attended | Reference group | | | | Reference group | | | | |
| Primary | **2.73 [1.55, 4.78]** | **0.001** | **0.004** | 2.33 [1.17, 3.47] | 1.33 [0.74, 2.40] | 0.336 | 0.504 | 1.29 [0.76, 2.11] | 1.79 [0.04, 3.54] |
| Secondary | **4.64 [3.91, 47.55]** | **0.001** | **0.004** | 3.30 [3.03, 8.41] | **6.00 [1.32, 27.36]** | **0.021** | **0.077** | **4.00 [1.28, 7.53]** | 5.23 [0.10, 10.36] |
| Tertiary | 1 | | | | 1 | | | | |
| Paternal education[m] | | | | | | | | | |
| Never attended | Reference group | | | | Reference group | | | | |
| Primary | 0.83 [0.37, 1.86] | 0.644 | 0.722 | 0.84 [0.40, 1.71] | 1.01 [0.45, 2.26] | 0.982 | 0.982 | 1.01 [0.48, 2.01] | 0.91 [0.02, 1.80] |
| Secondary | 2.13 [0.80, 5.56] | 0.128 | 0.211 | 1.91 [0.82, 3.82] | 1.55 [0.58, 4.12] | 0.380 | 0.518 | 1.47 [0.61, 3.14] | 1.79 [0.04, 3.55] |
| Tertiary | 3.52 [0.82, 15.06] | 0.090 | 0.169 | 2.81 [0.84, 6.30] | 6.00 [0.68, 52.90] | 0.107 | 0.214 | 4.00 [0.70, 8.55] | 4.44 [0.09, 8.79] |
| Number of children ever born: Median (Q1, Q3) | **0.84 [0.77, 0.93]** | **0.001** | **0.004[m]** | 0.85 [0.79, 0.94] | **0.78 [0.68, 0.90]** | **0.001** | **0.010[m]** | 0.80 [0.70, 0.91] | 0.81 [0.02, 1.60] |

*(Continued)*

| Risk factor variables | Kilifi Autism study | | | | NeuroDev Study | | | | OR Meta-analysis from the two studies |
|---|---|---|---|---|---|---|---|---|---|
| | Odds ratio [95% CI] | p-value | Adjusted p-value | Risk Ratio [95%CI] | Odds ratio [95% CI] | p-value | Adjusted p-value | Risk Ratio [95%CI] | |
| Birth order: Median (Q1, Q3) | **0.91 [0.82, 0.99]** | **0.046** | 0.108[m] | 0.92 [0.84, 0.99] | **0.85 [0.76, 0.95]** | **0.004** | **0.030[m]** | 0.86 [0.79, 0.96] | 0.88 [0.02, 1.74] |
| Medical complications during pregnancy (gestational hypertension, diabetes, eclampsia and maternal bleeding) | **3.17 [1.61, 6.23]** | **0.001** | **0.004[m]** | 2.61 [1.52, 4.09] | 2.79 [0.60, 12.93] | 0.190 | 0.356 | 2.37 [0.63, 5.90] | 2.97 [0.06, 5.88] |
| Infection during pregnancy (fever, malaria and other infections) | 0.99 [0.53, 1.86] | 0.983 | 0.983 | 0.99 [0.56, 1.71] | **5.31 [1.56, 18.11]** | **0.008** | **0.034[m]** | 3.71 [1.48, 6.68] | 1.67 [0.03, 3.30] |
| Drug misuse during pregnancy | 0.56 [0.27, 1.17] | 0.122 | 0.211[m] | 0.59 [0.29, 1.15] | **2.38 [1.56, 18.11]** | **0.008** | **0.034[m]** | 2.09 [1.48, 6.68] | 0.91 [0.02, 1.80] |
| Delivery place – home | **0.32 [0.18, 0.59]** | **0.001** | **0.004[m]** | 0.34 [0.20, 0.62] | **0.21 [0.10, 0.44]** | **<0.001** | **0.010[m]** | 0.23 [0.11, 0.47] | 0.25 [0.01, 0.50] |
| Labour and birth complications (induced labour and prolonged labour, PROM, umbilical cord complications and MOH) | **5.41 [1.84, 15.91]** | **0.002** | **0.007[m]** | 3.75 [1.70, 6.39] | **7.30 [2.17, 24.61]** | **0.001** | **0.010[m]** | 4.48 [1.94, 7.32] | 6.21 [0.12, 12.30] |
| Hypoxic ischaemic encephalopathy | **10.52 [4.04, 27.41]** | **0.001** | **0.004[m]** | 5.39 [3.10, 7.53] | 1.00 [0.99, 1.00] | 0.055 | 0.138[m] | 1.00 [0.99, 1.00] | 1.83 [0.04, 3.62] |
| Birth weight in kgs [Mean, SD] | **1.00 [0.99, 1.00]** | **0.001** | **0.004[m]** | 1.00 [0.99, 1.00] | 0.52 [0.27, 1.01] | 0.053 | 0.138[m] | 0.55 [0.29, 1.01] | 0.68 [0.01, 1.35] |
| Low birth weight [≤2.5 kg] | **0.76 [0.56, 1.04]** | 0.083 | 0.169[m] | 0.78 [0.59, 1.04] | **1.00 [0.99, 1.00]** | **0.042** | 0.126[m] | 1.00 [0.99, 1.00] | 0.86 [0.01, 1.35] |
| Previous hospitalisation in childhood | **6.41 [3.53, 11.62]** | **0.001** | **0.004[m]** | 4.16 [2.82, 5.64] | 1 [Collinear] | | | | N/A |
| Family history of seizures | 0.77 [0.35, 1.73] | 0.583 | 0.687 | **0.79 [0.37, 1.61]** | 1.24 [0.62, 2.48] | 0.546 | 0.665 | 1.21 [0.64, 2.16] | 0.86 [0.02, 1.71] |
| Neonatal jaundice | 2.14 [0.26, 17.50] | 0.477 | 0.630 | 1.92 [0.28, 6.60] | **5.49 [1.61, 18.72]** | **0.007** | **0.034[m]** | 3.79 [1.52, 6.75] | 3.08 [0.06, 6.10] |
| Seizures at birth | 6.15 [0.77, 49.19] | 0.087 | 0.169[m] | 4.06 [0.79, 8.45] | **4.23 [1.23, 14.57]** | **0.023** | 0.077[m] | **3.20 [1.20, 6.18]** | 5.01 [0.10, 9.92] |
| Family history of NDDs | Not assessed | | | | 1[Collinear] | | | | N/A |
| Cerebral malaria anytime in childhood | Not assessed | | | | **8.53 [1.11, 65.65]** | **0.039** | 0.114[m] | **4.87 [1.10, 8.79]** | N/A |
| Head injury | **3.67 [1.04, 12.95]** | **0.043** | 0.108[m] | 2.90 [1.04, 5.90] | Not assessed | | | | N/A |
| Malaria before 3 years | 1.32 [0.39, 4.49] | 0.656 | 0.722 | 1.28 [0.42, 3.33] | Not assessed | | | | N/A |

*(Continued)*

 

**Table 4.** (Continued)

| Risk factor variables | Kilifi Autism study | | | | NeuroDev Study | | | | OR Meta-analysis from the two studies |
|---|---|---|---|---|---|---|---|---|---|
| | Odds ratio [95% CI] | p-value | Adjusted p-value | Risk Ratio [95%CI] | Odds ratio [95% CI] | p-value | Adjusted p-value | Risk Ratio [95%CI | |
| Seizures before 3 years | 12.60 [2.95, 53.70] | 0.010 | 0.028[m] | 5.83 [2.47, 8.57] | Not assessed | | | | N/A |

Note: NDD = Neurodevelopmental Disorder, TD = Typically Developing, CI = Confidence Interval, Q1, Q3 = Quartile 1, Quartile 3, SD = standard deviation (mean and SD are provided for continuous variables with a normal distribution and median and Q1, Q3 are provided for count variables or continuous variables without normal distribution), SE- standard error, * adjusted p values after Benjamini Hochberg Procedure to control the false discovery rate at 0.05, p-values in **bold** <0.05, p-values in superscript

[m] ≤ 0.25 and included in multivariable analysis, effect sizes = meta-analysis of odds ratio from logistic regression.

## Risk factors

In the unadjusted univariable analysis, the occurrence of seizures before three years of age, HIE, labour and birth complications and previous hospitalisation in childhood were increased in the NDD group compared to the TD group. Maternal age (older) in the Kilifi Autism study and labour and birth complications, infection during pregnancy, neonatal jaundice, cerebral malaria anytime in childhood and previous hospitalisation in childhood in the NeuroDev study emerged as statistically significant risk factors associated with NDDs. The multivariable analysis identified medical complications during pregnancy, such as gestational hypertension and diabetes, eclampsia, maternal bleeding, HIE, and the mother's age in years, were associated with NDDs in the Kilifi Autism study. However, this effect was not observed in the NeuroDev study, where labour and birth complications were not significant risk factors in the Kilifi Autism study but were associated with autism in the NeuroDev study.

Medical complications during pregnancy, such as gestational hypertension, diabetes, and eclampsia, are complex problems in pregnancy that have been associated with autism and other neurodevelopmental conditions [48]. Biological pathways have been postulated, including a model of shallow placentation and an increase in placental debris in the maternal circulation, which, thereby, leads to a maternal immune response that affects placental, foetal circulatory systems and neurodevelopment [49]. Preeclampsia, for example, is moderated by other underlying factors such as maternal age and maternal cardiovascular and metabolic health [50].

Adverse perinatal events such as birth asphyxia are one of the leading causes of neonatal mortality in low and middle-income countries and are linked to intellectual disability, autism and cerebral palsy [51]. Studies have discussed hypoxic-ischemic damage, the prolonged or acute oxygen deprivation that may affect brain regions that are especially vulnerable to perinatal insults, which in turn contributes to inflammation and oxidative damage [52].

Maternal infection during pregnancy has been found to increase the risk of autism through the activation of the maternal immune response [53]. Population studies have singled out viral infections in the first trimester, bacterial infections in the second trimester, and influenza and febrile episodes in the entire pregnancy period, but especially in the third trimester [54,55]. Our findings appear consistent with the literature, as we found maternal infection during pregnancy to be a significant risk factor, with the effect persistent in the multivariable analysis for the NeuroDev study.

Prolonged labour was the most noted labour and birth complication in the NeuroDev study, with 22 mothers reporting this occurrence, major obstetric bleeding (n = 8), and three mothers also reported PROM. An American study found that children in the exogenous oxytocin drug-administered group were 2.32 times more likely to exhibit an autism phenotype. In contrast, another study in the US found that autism risk was not associated with oxytocin use alone, but when used with labour epidural analgesia [56]. There is concern that synthetic oxytocin administered during labour may enter the foetal circulation system, deactivate oxytocin receptors and disrupt the oxytocinergic system, thus increasing the risk for NDDs such as autism and ADHD [57,58]. While both positive and null findings of this association have been reported, a

systematic review in 2018 indicates that the link between synthetic oxytocin and NDDs is weak, with a call for further studies investigating oxytocin-stimulated labour and NDDs to account for maternal psychopathology and including sibling comparison in the study design [59]. Maternal age at birth also conferred a smaller increased risk. In the Kilifi Autism study, labour and birth complications were non- significant after the adjusted analysis (though they trended towards significance) but were significant in the NeuroDev study.

Malaria before three years of age was associated with autism in Tanzania [16], but we did not find this association significant in the adjusted analysis. Interestingly, we see a strong and persistent statistically significant association between seizures before age 3 years when we compared the NDD and typically developing children in both univariable and adjusted analyses. Malaria is one of the most common parasitic infections on the Kenyan coast and the most common cause of seizures in children under 5 years [60]. Brain MRI findings in a study in Uganda gave evidence for ischaemic neural injury upon exposure to cerebral malaria and other infections, which may be a pathway of interest in autism and other NDDs [61]. This requires further investigation.

Neonatal jaundice is associated with NDDs in the NeuroDev study. A population study in Denmark found that exposure to jaundice in neonates born at full term (≥38 weeks) was linked to an increased risk of developmental disorders [62]. A study in Egypt also found a similar association in a sample of 80 autistic children.[63] The impact of elevated serum bilirubin levels on neurodevelopment has been a significant concern in both clinical practice and scientific research [64]. A multisite case-control study in the US also found this association between jaundice and autism in neonates born between 35–37 weeks [65]. A systematic review of low-risk bias studies consolidated findings from 32 studies examining the association between neonatal jaundice and autism and found limited convincing evidence of an association between neonatal jaundice and autism [66].

Low birthweight and drug misuse emerged as important across the two datasets in the multivariable analysis, but the associations for the latter were conflicting between the datasets. Associations between substance use and NDDs are reported in other studies [67]. Intra-utero exposure to alcohol and tobacco is also thought to affect brain development and function through targets in the intrauterine environment [68]. The conflicting results for drug misuse across the datasets may be explained by spurious findings from the few observations and temporal imprecision of exposure in the Kilifi Autism study.

Interpretation of these risk factors individually may be difficult as the biological mechanisms of these factors are very likely closely related. For example, maternal bleeding during pregnancy is associated with foetal hypoxia, which is, in turn, related to the risk of NDDs [69]. Foetal distress, gestational hypertension, prolonged labour, and cord complications are also correlated with foetal hypoxia, and all these are associated with NDD risk [70]. The shared biological pathway for these factors is postulated to be oxygen deprivation during development [70]. Prenatal complications are hypothesized to occur as a result of autistic or neurodevelopmental conditions or a combinatory effect with genetic factors [71]. There are studies that have found that there are genetic factors associated with febrile seizures, infections such as meningitis and even neonatal jaundice [72]. Children with NDDs often will have mothers who have experienced more pregnancy and birth complications compared to their unaffected siblings and typically developing children. Recent research has also focused on the foetus' involvement in the birth process, with studies finding that the foetus actively engages in the birth process through various physiological mechanisms [73]. Hypotheses attempt to explain this observation. Two competing hypotheses include that NDDs such as autism are aetiologically heterogeneous due to genetic and environmental factors such as birth and labour complications, the second hypothesis is that genetic and familial factors increase the risk of both autism and birth and labour complications, as evidenced by findings in conditions such as Prader-Willi syndrome and Down syndrome [74,75]. Researchers also discuss a third hypothesis: in the presence of gene-environment interactions, birth and labour complications may play a role in autism and NDDs, with the modulation of environmental risk factors due to familial liability as seen in meta-analysis of twin studies [76]. These studies highlight the interaction between genetic, familial and environmental factors, with more studies needed using various family structures such as twin studies, and studies of non-affected family members to investigate this interplay.

Sex and preterm birth were not statistically significant replicated risk factors associated with autism after the adjusted multivariable analysis in this study. Maternal age confers a slight increase in odds of NDDs in the Kilifi Autism study but not in the NeuroDev study. A meta-analysis by Wu et al. found that in approximately 30 studies, there was an increased risk of approximately 40% and 50% for the oldest maternal and paternal age categories [77]. Studies have also shown increases in risk for maternal age above 35 years and above paternal age of 40 years [78]. Plausible suggestions have been made regarding the combined effect of maternal and paternal age, though the mechanisms of these changes are not as well elucidated. Potential explanations include epigenetic modification and DNA damage due to ageing and mediation due to pregnancy complications [79].

Maternal education, one of the parental factors that was adjusted for in the multivariable analysis, is important to investigate fully. Studies have found that parental educational level and other sociodemographic markers may affect the awareness of NDDs and help-seeking behaviour due to concerns about their child's development [80]. In our analysis, maternal education was associated with delivery place; we found that mothers with higher education levels gave birth in hospitals as opposed to giving birth at home. We also found an association between maternal education level and NDD diagnosis, with mothers of children with an NDD diagnosis having higher education levels, supporting the link between maternal education and developmental concerns.

## Protective factors

Higher parity (the more children a mother had) and child age (the child being older) at assessment seemed to confer a protective effect when we compared the NDD and TD groups. The association of earlier birth order and reduced risk for NDD is consistent with findings from another study, supporting the findings on the role of birth order in autism (discussed in the discovery dataset above) [81]. This is likely because of reproductive stoppage, whereby parents of children with NDDs might pause or stop expanding the family after having a child with an NDD [82].

Even with the adjusted analysis, delivery at home still was associated with the least risk (OR=0.31 [95%CI 0.12, 0.80]). The Kenyan government has increased efforts in the past few decades to increase hospital access to mothers with the introduction of the Free Maternity Policy in 2013, and in 2014, 52.3% of women in Kilifi County gave birth at a health facility [83]. For mothers giving birth at home, long distance from a healthcare facility is one main reason cited, which is compounded by limited access to transport services and poor road networks [84]. The association observed with delivery at home is not causal, and with a little more disentangling, it may suggest that mothers of children with NDDs tend to have higher education levels and may be aware of the potential of obstetric complications during birth, leading them to give birth in a hospital setting. Another explanation is that with limited resources and access to health care, mothers who had a pregnancy with fewer complications may have considered themselves unlikely to have complications during labour and birth and, as such, would more readily give birth at home as opposed to those with pregnancy complications, who would likely present themselves to hospital for labour and birth as they perceived themselves as "at risk" [85].

## Strengths and limitations of the study

With this study, we add much-needed data regarding environmental factors associated with NDDs in countries on the African continent, and with the use of two independent datasets, we were able to evaluate the replication of the risk factors with another dataset. There are some limitations, particularly with the reliance on parental reports, which might lead to some recall bias as opposed to precise measurement data on health cards and medical reports. There is a lot of missing data, such as APGAR scores and accurate birth weight. This may have led to over or under-reporting of certain risk factors such as HIE. This may lead to the true link between these factors and NDDs being underestimated or overestimated, leading to biased associations.

The Kilifi Autism study and the NeuroDev study were conducted at different times, which may introduce temporal biases. Changes in healthcare access and environmental factors over time could affect the comparability of the data. This temporal discrepancy might influence the observed associations between environmental exposures and NDDs. The criteria and processes used to diagnose and classify NDD participants vary slightly between the Kilifi Autism study and the NeuroDev study. This may have led to some slight discrepancies in comparison between NDD participants from both studies; however, to mitigate this we combined the specific NDD diagnoses to a more agreeable definition of case status as a diagnosis of an NDD.

Because of our modest sample size, we were not able to interrogate specific risk factors, which were grouped into another variable, for example, hypertensive disorders, eclampsia and maternal bleeding, which were grouped into medical complications during pregnancy. As such, the subtleties in the association between these specific risk factors and the outcome may be undetected or there may have been a bias in the estimates reported.

It would have been useful to investigate risk factors such as maternal mental health and general parental psychiatric history, which is a significant confounding factor for perinatal risk factors and NDDs. Studies have found that both maternal and parental prenatal mood and anxiety disorders, maternal eating disorders, and exposure to prenatal stress are significantly linked to autism and ADHD [86–88]. We were also unable to incorporate genetic results and other potential biomarkers, such as neuroimaging. The addition of genetic approaches would provide deeper insights into the biological mechanisms of NDDs and potentially evaluate gene-environment interactions while the incorporation of neuroimaging would reveal structural, functional or connectivity differences in the brain associated with NDDs. We were also missing information on other biological measures of infection and inflammation in pregnancy and nutritional factors, such as iron deficiency in pregnancy and early childhood. These factors would also help identify pathways linking or disruptions of critical pathways, for example, with maternal immune activation, to neurodevelopmental outcomes. Finally, there is also an opportunity to investigate risk and protective factors in prospective study designs, including younger children, to better understand factors that can connect brain and behaviour change and interventions due to these modifiable factors [89].

## Conclusion and clinical and research implications

In conclusion, the study identified labour and birth complications, HIE, neonatal jaundice and seizures before age 3 years as pervasive significant risk factors associated with NDDs. These factors were significant in either the NeuroDev or Kilifi Autism studies, not both. Further studies looking into the aetiology of NDDs, particularly ones that examine genetic and environmental interaction, are encouraged. In the NeuroDev study, we will have the opportunity to incorporate some genetic findings in our further evaluation of risk factors. Recognition of prenatal, perinatal and childhood risk factors is important clinically and in research as these factors hold the promise of guiding developmental screening and monitoring to aid in earlier identification and screening of autism, thereby reducing the delay in diagnosis and aiding quicker diagnosis and referral to care.

## Supporting information

**S1 Table. Adjusted multivariable analysis of relevant parental, perinatal and neonatal factors associated with autism after adjusting for parental socioeconomic and demographic status –NDD vs TD group.**
(DOCX)

## Acknowledgments

We express our immense gratitude to the Kilifi Autism Study and NeuroDev Kenya participants who volunteered to participate in these studies. We would also like to thank all the study staff in KEMRI-Wellcome Trust involved in the study.

## Author contributions

**Conceptualization:** Kirsten A. Donald, Amina Abubakar, Charles R. Newton.

**Data curation:** Patricia Kipkemoi, Paul Mwangi, Collins Kipkoech.

**Formal analysis:** Patricia Kipkemoi, Jeanne E. Savage.

**Funding acquisition:** Kirsten A. Donald, Elise Robinson, Amina Abubakar, Charles R. Newton.

**Investigation:** Patricia Kipkemoi, Joseph Gona, Kenneth Rimba, Martha Kombe, Alfred Ngombo, Beatrice Mkubwa, Constance Rehema.

**Methodology:** Patricia Kipkemoi, Jeanne E. Savage, Amina Abubakar, Charles R. Newton.

**Project administration:** Eunice Chepkemoi.

**Resources:** Amina Abubakar, Charles R. Newton.

**Supervision:** Jeanne E. Savage, Danielle Posthuma, Amina Abubakar, Charles R. Newton.

**Validation:** Jeanne E. Savage.

**Writing – original draft:** Patricia Kipkemoi.

**Writing – review & editing:** Jeanne E. Savage, Joseph Gona, Kenneth Rimba, Paul Mwangi, Collins Kipkoech, Eunice Chepkemoi, Martha Kombe, Alfred Ngombo, Beatrice Mkubwa, Constance Rehema, Symon M. Kariuki, Danielle Posthuma, Kirsten A. Donald, Elise Robinson, Amina Abubakar, Charles R. Newton.

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
