## [Decision Letter · Decision Letter 0]

21 Nov 2024

PGPH-D-24-02176

Socio-medical Factors Associated with Neurodevelopmental Disorders on the Kenyan Coast

Dear Dr. Kipkemoi,

Thank you for submitting your manuscript to PLOS Global Public Health. After careful consideration, we feel that it has merit but does not fully meet PLOS Global Public Health’s publication criteria as it currently stands. Therefore, we invite you to submit a revised version of the manuscript that addresses the points raised during the review process.

We look forward to receiving your revised manuscript.

Kind regards,

Massimiliano Orri, PhD

Academic Editor

Journal Requirements:

**Please only choose the relevant sentences from below**

1. Please clarify all sources of funding (financial or material support) for your study. List the grants (with grant number) or organizations (with url) that supported your study, including funding received from your institution. 

2. State the initials, alongside each funding source, of each author to receive each grant.

3. State what role the funders took in the study. If the funders had no role in your study, please state: “The funders had no role in study design, data collection and analysis, decision to publish, or preparation of the manuscript.”

4. If any authors received a salary from any of your funders, please state which authors and which funders.

2. Please send a completed 'Competing Interests' statement, including any COIs declared by your co-authors. If you have no competing interests to declare, please state "The authors have declared that no competing interests exist". Otherwise please declare all competing interests beginning with the statement "I have read the journal's policy and the authors of this manuscript have the following competing interests:"

3. In the online submission form, you indicated that "Coded individual-level data that do not allow researchers to identify participants will be made available by the authors, without undue reservation. Curated data for this manuscript, will be deposited to the Harvard Dataverse.". 

3. Uploaded as supplementary information.

4. Please provide separate figure files in .tif or .eps format.

5. Please provide an Author Summary. This should appear in your manuscript between the Abstract (if applicable) and the Introduction, and should be 150–200 words long. The aim should be to make your findings accessible to a wide audience that includes both scientists and non-scientists. Sample summaries can be found on our website under Submission Guidelines: 

https://journals.plos.org/globalpublichealth/s/submission-guidelines#loc-parts-of-a-submission

Additional Editor Comments (if provided):

Reviewers' comments:

Reviewer's Responses to Questions

**Comments to the Author**

1. Does this manuscript meet PLOS Global Public Health’s publication criteria ? Is the manuscript technically sound, and do the data support the conclusions? The manuscript must describe methodologically and ethically rigorous research with conclusions that are appropriately drawn based on the data presented.

Reviewer #1: Yes

Reviewer #2: Partly

Reviewer #3: Yes

2. Has the statistical analysis been performed appropriately and rigorously?

Reviewer #1: Yes

Reviewer #2: I don't know

Reviewer #3: Yes

3. Have the authors made all data underlying the findings in their manuscript fully available (please refer to the Data Availability Statement at the start of the manuscript PDF file)?

Reviewer #1: Yes

Reviewer #2: No

Reviewer #3: Yes

4. Is the manuscript presented in an intelligible fashion and written in standard English?

Reviewer #1: Yes

Reviewer #2: Yes

Reviewer #3: Yes

5. Review Comments to the Author

Reviewer #1: 1. The manuscript described the data and conclusions appropriately which was methodologically and ethically acceptable.

2. The statistical analysis was sophisticated and well presented

3. I'm satisfied about the availability of all data.

4. Manuscript was written in clear, grammatically correct standard English.

5. I have no concerns around ethical clearance

It would be better if the study was done in a younger age group of children. I have noted that the mean age of Kilifi study was 12 years and the TD group had the mean age was 9 years. I understand that SCQ lifetime was used, but exposure to developmental trauma is a challenge to the diagnosis of ASD in older children.

To explore etiology in ASD, I think it's better to assess children only with ASD without intellectual disability in your future studies.

Reviewer #2: Dear authors,

Thank you for the opportunity to review your manuscript, which addresses an important gap in the literature. I read with interest and hope that my feedback is useful in strengthening the manuscript for publication. Please see my feedback in attached letter to the editors.

Warm regards,

Lexy

Reviewer #3: 1. Summary of the research and overall impression

The manuscript presents the results of original research into the socio-medical factors associated with neurodevelopmental disorders in Kenya. This is a neglected area of public health particularly on the African continent. The authors demonstrated a good grasp of the relevant literature which was used to set the scene in relation to the burden of neurodevelopmental disorders. They outlined the peculiarity of the African socio-medical context, justifying the need to generate valid local epidemiological data to inform policymaking, practice and research.

The methods were adequately described, and the conclusions are appropriately supported by the data presented.

Overall, the authors engaged well with the topic in the conduct of the research and reporting of the study results and meet criteria for publication in PLOS Global Public Health.

2. Discussion of specific areas for improvement

While the manuscript is highly commendable for the above reasons, I have highlighted areas for improvement by addressing some minor issues:

1. Introduction: The introduction section is very thorough, the authors demonstrating a great understanding of the global and local landscape of neurodevelopmental conditions. However, I have made the following observations:

- The point made in Line 78, “The same may be true for the prevalence of autism in Africa which is thought to be similar to the global estimate or possibly greater given the high incidence of neuro-behavioural disorders such as adverse perinatal events, infections of the brain and environmental toxins” appears to be a repetition of the point already made in Line 74. Perhaps this should be deleted.

- For the statement made in Line 91, “This has changed little in the last decade”, it will be helpful to provide a reference if available.

2. Results:

- In Line 293, … “and for the NeuroDev study, 151 children with an NDD and 73 typically developing controls with a median age of 11 years (9,13)”, typically developing children included as controls was n=73 but it is n=71 in Table 2. The authors need to correct this.

- In Line 296, the authors state, “In the Kilifi Autism study, 41.8% of mothers did not have any formal schooling and 42.4% in the NeuroDev study, in comparison to 13.2% of fathers and 14.8% of fathers in the two studies, respectively” The proportions reported on Table 2 are different: 41.8% of mothers did not have any formal schooling in the Kilifi Autism Study and 42.4% in the NeuroDev study. The authors need to check and reconcile this.

- In Line 298, similarly, the proportions of fathers without any formal schooling the Kilifi Autism Study and the NeuroDev studie are 3.2% of fathers and 14.8% of fathers in the two studies, respectively. These proportions are different in Table 2: These are 9.3% and 14.0% of fathers in the Kilifi Autism Study and the NeuroDev study respectively The authors should also check and reconcile these.

- In Line 306, the statement, “For the NeuroDev study, we see that the ages of the NDD group mothers are 26 years old, and mothers of controls have a median age of 29 years” was already made earlier in Line 303. The authors should delete the second statement.

3. Discussion

- In Line 403, the statement, “Maternal age (older) in the Kilifi Autism study and labour and birth complications, infection during pregnancy, neonatal jaundice, cerebral malaria anytime in childhood and previous hospitalisation in childhood emerged as statistically significant risk factors associated with autism in the NeuroDev study” based on the results presented, should read, “Maternal age (older) in the Kilifi Autism study and labour and birth complications, infection during pregnancy, neonatal jaundice, cerebral malaria anytime in childhood and previous hospitalisation in childhood in the NeuroDev study, emerged as statistically significant risk factors associated with autism.

- Line 492: the space should be deleted.

- Statements on Line 511, “The protective effect observed with delivery at home in our study warrants more disentangling, as it may suggest that mothers of children with NDDs tend to have higher education levels and may be aware of the potential of obstetric complications during birth, leading them to give birth in a hospital setting” is essentially the same as the statement in Line 514. The authors should reconcile this.

- In Line 524, “The association of earlier order of birth order and reduced risk for NDD is consistent with findings from another study..” should read, “ The association of earlier birth order and reduced risk for NDD is consistent with findings from another study..

6. PLOS authors have the option to publish the peer review history of their article (what does this mean? ). If published, this will include your full peer review and any attached files.

**Do you want your identity to be public for this peer review?** For information about this choice, including consent withdrawal, please see our Privacy Policy .

Reviewer #1: No

Reviewer #2: **Yes: ** Lexy Staniland

Reviewer #3: **Yes: ** Muftau Mohammed

---

## [Decision Letter · Decision Letter 1]

17 Apr 2025

Socio-medical Factors Associated with Neurodevelopmental Disorders on the Kenyan Coast

PGPH-D-24-02176R1

Dear Ms Kipkemoi,

We are pleased to inform you that your manuscript 'Socio-medical Factors Associated with Neurodevelopmental Disorders on the Kenyan Coast' has been provisionally accepted for publication in PLOS Global Public Health.

Best regards,

Massimiliano Orri, PhD

Academic Editor

Reviewer Comments (if any, and for reference):

Reviewer's Responses to Questions

**Comments to the Author**

1. If the authors have adequately addressed your comments raised in a previous round of review and you feel that this manuscript is now acceptable for publication, you may indicate that here to bypass the “Comments to the Author” section, enter your conflict of interest statement in the “Confidential to Editor” section, and submit your "Accept" recommendation.

Reviewer #1: All comments have been addressed

Reviewer #2: All comments have been addressed

2. Does this manuscript meet PLOS Global Public Health’s publication criteria ? Is the manuscript technically sound, and do the data support the conclusions? The manuscript must describe methodologically and ethically rigorous research with conclusions that are appropriately drawn based on the data presented.

Reviewer #1: Yes

Reviewer #2: Yes

3. Has the statistical analysis been performed appropriately and rigorously?

Reviewer #1: Yes

Reviewer #2: Yes

4. Have the authors made all data underlying the findings in their manuscript fully available (please refer to the Data Availability Statement at the start of the manuscript PDF file)?

Reviewer #1: Yes

Reviewer #2: Yes

5. Is the manuscript presented in an intelligible fashion and written in standard English?

Reviewer #1: Yes

Reviewer #2: Yes

6. Review Comments to the Author

Reviewer #1: Author addressed my previous comment on importance of early diagnosis of ASD, because the samples had older children.

Reviewer #2: Dear authors,

Thank you for your careful consideration of and detailed responses to my feedback and suggestions. I am satisfied with the revisions made and commend you on a great manuscript!

Best regards,

Lexy

7. PLOS authors have the option to publish the peer review history of their article (what does this mean? ). If published, this will include your full peer review and any attached files.

**Do you want your identity to be public for this peer review?** For information about this choice, including consent withdrawal, please see our Privacy Policy .

Reviewer #1: **Yes: ** Darshani Rupasinghe

Reviewer #2: **Yes: ** Lexy Staniland
